# Clinical Relevance of Tumour-Infiltrating Immune Cells in HER2-Negative Breast Cancer Treated with Neoadjuvant Therapy

**DOI:** 10.3390/ijms25052627

**Published:** 2024-02-23

**Authors:** Cristina Arqueros, Alberto Gallardo, Silvia Vidal, Rubén Osuna-Gómez, Ariadna Tibau, Olga Lidia Bell, Teresa Ramón y Cajal, Enrique Lerma, Bárbara Lobato-Delgado, Juliana Salazar, Agustí Barnadas

**Affiliations:** 1Department of Medical Oncology, Hospital de la Santa Creu i Sant Pau, 08041 Barcelona, Spain; carqueros@santpau.cat (C.A.); atibau@santpau.cat (A.T.); tramon@santpau.cat (T.R.y.C.); 2Department of Medicine, Faculty of Medicine, Universitat Autònoma de Barcelona, 08035 Barcelona, Spain; 3Department of Pathology, Hospital de la Santa Creu i Sant Pau, 08041 Barcelona, Spain; agallardoa@santpau.cat (A.G.); elerma@santpau.cat (E.L.); 4Department of Morphological Sciences, Faculty of Medicine, Universitat Autònoma de Barcelona, 08035 Barcelona, Spain; 5Institut d’Investigació Biomèdica Sant Pau (IIB-Sant Pau), Institut de Recerca Sant Pau-CERCA Center, 08041 Barcelona, Spain; 6Inflammatory Diseases, Institut d’Investigació Biomèdica Sant Pau (IIB-Sant Pau), Institut de Recerca Sant Pau-CERCA Center, 08041 Barcelona, Spain; svidal@santpau.cat (S.V.); rosuna@santpau.cat (R.O.-G.); 7Translational Medical Oncology Laboratory, Institut d’Investigació Biomèdica Sant Pau (IIB-Sant Pau), Institut de Recerca Sant Pau-CERCA Center, 08041 Barcelona, Spain; obell@santpau.cat; 8Unitat de Genòmica de Malalties Complexes, Institut d’Investigació Biomèdica Sant Pau (IIB-Sant Pau), Institut de Recerca de l’Hospital de la Santa Creu i Sant Pau-CERCA Center, 08041 Barcelona, Spain; blobato@santpau.cat; 9Centro de Investigación Biomedica en Red Cancer (CIBERONC), Instituto de Salud Carlos III, 28029 Madrid, Spain

**Keywords:** breast cancer, neoadjuvant chemotherapy, stromal tumour-infiltrating lymphocytes, tumour-infiltrating immune cells

## Abstract

Currently, therapy response cannot be accurately predicted in HER2-negative breast cancer (BC). Measuring stromal tumour-infiltrating lymphocytes (sTILs) and mediators of the tumour microenvironment and characterizing tumour-infiltrating immune cells (TIICs) may improve treatment response in the neoadjuvant setting. Tumour tissue and peripheral blood samples were retrospectively collected from 118 patients, and sTILs were evaluated. Circulating exosomes and myeloid-derived suppressor cells were determined by flow cytometry. TIICs markers (CD4, CD8, CD20, CD1a, and CD68) were assessed immunohistochemically. High sTILs were significantly associated with pathological complete response (pCR; *p* = 0.048) and event-free survival (EFS; *p* = 0.027). High-CD68 cells were significantly associated with pCR in triple-negative (TN, *p* = 0.027) and high-CD1a cells with EFS in luminal-B (*p* = 0.012) BC. Cluster analyses of TIICs revealed two groups of tumours (C1 and C2) that had different immune patterns and clinical outcomes. An immunoscore based on clinicopathological variables was developed to identify high risk (C1) or low-risk (C2) patients. Additionally, cluster analyses revealed two groups of tumours for both luminal-B and TNBC. Our findings support the association of sTILs with pCR and show an immunological component in a subset of patients with HER2-negative BC. Our immunoscore may be useful for future escalation or de-escalation treatments.

## 1. Introduction

Breast cancer (BC) is characterized by a heterogeneous histological origin and composition that influences treatment response and disease survival [1]. Thus, therapy response cannot be precisely predicted only by current clinical and pathological parameters, and a better understanding of tumour biology and its microenvironment is needed. The tumour microenvironment has a critical role in tumour differentiation, tumour progression, and treatment response [2], and its characterisation could improve molecular classification and the selection of more personalized treatment regimens.

Assessment of tumour-infiltrating immune cells (TIICs) has shown clinical promise in predicting BC outcomes. In HER2-positive and triple-negative (TN) BC patients treated with neoadjuvant chemotherapy (NACT), higher stromal tumour-infiltrating lymphocytes (sTILs) have been associated with a better prognosis [2,3] and a higher probability of achieving pathological complete response (pCR) [4]. To date, the premises of the TILs working group [5] have not been used in the selection of treatments, probably due to a lack of standardisation of the method. Moreover, growing evidence suggests that subpopulations of TIICs could also be determinants of BC outcomes [5,6,7], although it has not been established which subpopulations are most informative. For instance, CD8 cytotoxic T lymphocytes [8,9] and CD4 Th1 cells [10] have been associated with prognosis regardless of BC subtype.

Tumour immunity involves complex interactions between antitumoural and protumoural immune cell mediators, including sTILs, myeloid-derived suppressor cells (MDSCs) [11], and exosomes, among others [12]. High expression of tumour-infiltrating MDSCs has been associated with a low pCR rate and poor prognosis in BC [13,14]. Exosomes are considered potential non-invasive biomarkers for the diagnosis, prognosis, and therapeutics of several diseases, including BC [15,16].

The molecular classification of BC has improved with the determination of gene expression [17,18], but predicting individual prognosis remains a challenge. Evidence supports sTILs as a prognostic marker [3,19,20,21] and as a predictor of response in NACT [22] in TNBC and HER2-positive subtypes, but the utility of sTILs in the luminal HER2-negative subtype remains uncertain, and few studies provide separate data for luminal-A and luminal-B subtypes [4,23,24,25]. Further characterisation of TIICs and their value in predicting outcomes is needed [26]. The aims of this study were to analyse the association of sTILs with pathological response and survival and with other systemic biomarkers, and to develop a new risk score to predict outcome from the expression of TIICs in a cohort of HER2-negative BC treated with NACT.

## 2. Results

### 2.1. Patients’ Characteristics

Table 1 summarises the baseline clinicopathological characteristics of the participants. The patients’ median age at diagnosis was 55.1 (27–86) years, and the median follow-up was 5.1 (0.8–9.6) years. During the inclusion period, 10 (9%) patients had a local recurrence or distant progression, and 4 (3%) patients died. Ninety-one (77%) of the patients were classified as having luminal tumours (37 luminal-A and 54 luminal-B) and 27 (23%) as having TNBC tumours. A pCR was achieved in 14% of the patients after neoadjuvant treatment (8 luminal and 8 TNBC).

### 2.2. sTILs and Clinicopathological Features

All 118 tumour samples were assessed for sTILs levels. Fifty-three presented high sTILs (37 luminal and 16 TN), and 65 low-sTILs (54 luminal and 11 TNBC) according to the predefined cut-off. sTILs were statistically associated with histological grade (mean sTILs G1/G2: 19.8 (25.9) vs. G3: 33.9 (35.6); OR, 0.86; 95% CI, 0.76–0.98; *p* = 0.023) (Figure 1A) (Appendix A). Differences were found between TNBC and luminal tumours, although they were not significant (*p* = 0.06) (Figure 1B). We also observed numerical differences for sTILs as a stratified variable; 37.7% of the tumours with high sTILs were G3 compared to 21.5% with low-sTILs (OR, 0.45; 95% CI, 0.20–1.02; *p* = 0.058) (Appendix A).

### 2.3. sTILs as Predictive Biomarkers

sTILs were significantly associated with pCR, both as a continuous variable (mean sTILs responders: 42.5 (38.8) vs. non-responders: 21.0 (26.9); *p* = 0.010) and as a stratified variable (responders high sTILs: 12/53, 22.6% vs. low-sTILs: 4/65, 6.15%; *p* = 0.011) (Table 2). These differences were maintained in the multivariate analyses after adjusting for histological grade, tumour size, lymph node status, and molecular subtype (*p* = 0.046 and *p* = 0.048, respectively). We also observed numerical differences in the luminal subtype (responders high sTILs: 16.2% vs. low-sTILs: 3.7%; univariate *p* = 0.058).

sTILs were also significantly associated with a pathological response by MP in the breast, as a continuous variable (mean sTILs responders: 34.9 (36.0) vs. non-responders: 19.5 (25.5); *p* = 0.013) and as a stratified variable (responders high sTILs: 21/53, 39.6% vs. low-sTILs: 13/65, 20%; *p* = 0.022) (Table 2). These significances were maintained in the multivariate analyses after adjusting for histological grade, tumour size, lymph node status, and molecular subtype (*p* = 0.027 and *p* = 0.045, respectively). We also obtained significant associations for the luminal subtype (high sTILs responders: 35.1% vs. low-sTILs responders: 16.7%; univariate *p* = 0.043).

### 2.4. sTILs as Prognostic Biomarker

We found a significant association between sTILs as a stratified variable and EFS (univariate *p* = 0.055 and multivariate *p* = 0.027) (Table 3). The 5-year EFS was 98.1% (95% CI: 87.1–99.7) for high sTILs compared to 89.8% (95% CI 78.4–95.3%) for low-sTILs. For TNBC, the risk of relapse was higher in patients with low-sTILs than in those with high sTILs (*p* = 0.024). Similar trends were observed for the luminal subtype, but they were not significant (*p* = 0.171). We found non-significant associations between sTILs and OS.

### 2.5. Systemic Expression Biomarkers

sTILs as a continuous variable were inversely associated with exosomes (r = −0.293; *p* = 0.035) (Figure 2A) and MDSCs (r = −0.277; *p* = 0.039) (Figure 2B). sTILs as a stratified variable was also associated with MDSCs (*p* = 0.031) (Appendix A). We found similar results for the luminal subtype (Appendix A). None of the systemic biomarkers were associated with pathological response or survival.

### 2.6. TIICs and Outcomes

High-CD68 levels were significantly associated with pCR in TNBC (responders high-CD68: 7/16, 43.8% vs. low-CD68: 0/9, 0%; *p* = 0.027). We also found numerical differences between high-CD4 and pathological response according to MP in the breast, in the entire cohort (responders high-CD4: 19/47, 40.4% vs. low-CD4: 12/53, 22.6%; *p* = 0.055), and in luminal-B tumours (responders high-CD4: 9/19, 47.4% vs. low-CD4: 5/25, 20%; *p* = 0.054). High-CD1a levels were significantly associated with worse EFS (*p* = 0.012), and high-CD4 showed numerical differences with better EFS (*p* = 0.06) in luminal-B tumours. We found no association of TIICs with OS.

### 2.7. Clustering TIICs

We performed unsupervised hierarchical clustering analysis based on the expression pattern of the TIICs markers (CD20, CD8, CD4, and CD68). Patients’ tumours clustered into C1 and C2 (Figure 3).

We found significant associations with histological grade: 52.4% (11/21) of the tumours in C2 were G3 compared to 22.8% (18/79) in C1 (*p* = 0.012). We also found numerical differences with the rate of disease proliferation: 81% (17/21) of the tumours within C2 had ki67 > 14% compared to 58.2% (46/79) within C1 (*p* = 0.057). sTILs were significantly associated with the clusters both as a continuous variable (mean sTILs C1: 18.5 (24.6) vs. C2: 53.6 (34.9); *p* < 0.001) and as a stratified variable (C2 high sTILs: 18/47, 38.3% vs. low-sTILs: 3/49, 6.12%; *p* < 0.001) (Table 4). Patients in C2 achieved better clinical results than patients in C1. The difference in pathological response assessed according to MP in the breast was statistically significant (univariate *p* = 0.024 and multivariate *p* = 0.057) (Table 5).

We also identified two clusters for luminal-B tumours, C3 and C4 (Figure 4A). Patients in C4 achieved better clinical results than patients in C3. sTILs were significantly associated with the clusters as a continuous variable (mean sTILs C3: 19.6 (23.1) vs. C4: 43.6 (35.1); *p* = 0.028) (Table 4). The differences in tumour response were statistically significant when assessed according to MP in the breast (univariate *p* = 0.046 and multivariate *p* = 0.023) and in the axilla (univariate *p* = 0.039 and multivariate *p* = 0.020) (Table 5). For TNBC, we identified C5 and C6 clusters (Figure 4B), but there were no statistical differences.

We did not identify any significant associations with survival.

### 2.8. Immunoscore

We created a predictive model to obtain a risk score to categorise patients as C1 or C2. The weight for the risk score of the variables that fitted the model was 1 for age ≥ 50 (OR, 2.13; 95% CI, 0.70–6.51; *p* = 0.186), 1 for histological grade G1/G2 (OR, 2.83; 95% CI, 0.93–8.61; *p* = 0.066), 2 for low-sTILs (OR, 8.39; 95% CI, 2.21–31.84; *p* = 0.002), and 0 for age < 50, histological grade G3, and high sTILs. The simplified predictive model for the area under the ROC was 0.8079. Score values were 0 to 1 for low-risk (C2) and 2 to 4 for high risk (C1).

## 3. Discussion

The tumour microenvironment can affect response to systemic therapy and may help identify patients at high risk of relapse who may benefit from treatment escalation or de-escalation in the neoadjuvant setting [27]. Here we provide additional evidence of the predictive value of sTILs in HER2-negative BC and describe associations of sTILs found with circulating mediators of the tumour microenvironment. Moreover, we propose a novel immunoscore based on TIICs that may help select treatment strategies.

sTILs are immune cells that migrate to a malignant lesion as part of an immune response. Denkert et al. [4] confirmed previous findings [22,28,29,30] showing an association between high sTILs and pCR in NACT in BC, regardless of the molecular subtype. They also observed an association with better prognosis, which was measured as disease-free survival (DFS) in patients with HER2-positive and TNBC tumours but not in patients with luminal-HER2-negative tumours. Surprisingly, low-sTILs were associated with higher OS in the luminal subgroup. However, other studies failed to find a benefit in DFS for high sTILs in luminal tumours [7,31,32].

We found that patients with high sTILs were more likely to achieve a pathological response. Although this could have had an impact on EFS, we did not observe this association in OS, probably due to the low number of events and the need for a longer follow-up period. For TNBC, we found that patients with low-sTILs had worse EFS, although we did not observe an impact on pathological response. These results are in line with previous studies showing an association between high sTILs and a good prognosis in patients with TNBC [4,7]. Interestingly, we observed similar results for luminal BC. We found an association between high sTILs and pCR and better EFS, although the latter was not statistically significant. Our results are consistent with those of Curigliano et al. [23]. They found that high sTILs (≥5%) were associated with higher distant DFS in high risk patients treated with adjuvant chemotherapy, but not, in keeping with other studies [33], in untreated patients. We suggest that high sTILs impact positively on survival, especially in high risk patients with luminal tumours, and that there is a more immunogenic subgroup within the luminal tumours that deserves further exploration as they could be candidates for immunotherapy.

To better understand tumour immune evasion mechanisms, we analysed other immune cells and mediators determined post-NACT and observed weak negative correlations between expression of sTILs and circulating exosomes and MDSCs in the overall population and in the luminal type. However, these results need to be validated both before and after NACT before drawing conclusions. Most studies on BC have reported that exosomes and MDSCs have a suppressive effect on the antitumour immune response by inhibiting T cells [34,35], but a dynamic exploration measuring the immune response at baseline and after chemotherapy is warranted [14,36].

When we evaluated the most representative TIICs individually, we found a correlation between pathological response and high-CD4 and high-CD68 cells, the latter having been previously described in TNBC [37,38]. Moreover, we found that intratumourally high-CD1a cells were associated with worse EFS in the luminal-B subtype. High CD1a levels have been associated with worse outcomes in TNBC [39,40,41], but no data are available in luminal tumours.

Because the interaction between immune cell populations may be important in the tumour response to therapy, we performed cluster analysis of TIIC expression and identified two groups of tumours. One group (C2) expressed pro-inflammatory cells involving macrophages, B cells, and T cells and was related to a higher histological grade and rate of disease, higher sTILs, a better pathological response, and a lack of relapses. The other group (C1) presented under-expression of the inflammatory infiltrate and was related to worse outcomes. These data revealed which cell populations may be involved in clinical outcomes. The predictive value of CD4 and CD8 T lymphocytes was in concordance with other studies [10,42], and the role of macrophages and B cells may be similar to that of CD4 and CD8 T lymphocytes [42,43,44].

Our findings suggest that classifying tumours according to the expression of TIICs may improve therapeutic management in HER2-negative BC patients. The C2 group included low-risk patients who could be eligible for de-escalation therapy or candidates for immunotherapy. In contrast, the C1 group included high risk patients who could benefit from NACT escalation strategies, such as those stimulating the immune response. Therefore, to facilitate the implementation of this classification, we propose an immunoscore based on clinicopathological characteristics that groups patients into high risk (C1) or low-risk (C2) groups, thereby avoiding the need for immunohistochemistry.

We also performed cluster analysis for each molecular subtype to better understand the highly heterogeneous HER2-negative BC tumours. We identified a group of tumours for luminal-B (C4) and another for TNBC (C6) that achieved better clinical results. These groups were also characterized by an enrichment of histiocytes and T and B lymphocytes, such as those classified in C2. In the luminal context, our results are in line with those supporting activation of an immunogenic response in certain luminal tumours [45].

Our study provides additional data on the involvement of TIICs in the outcomes of patients with HER2-negative BC receiving NACT, particularly in the luminal type. There are several limitations, however. First, our sample size was small, and the follow-up period was short. Second, baseline blood samples were not available for systemic biomarker analyses, and post-NACT blood samples were not available from all the patients because of the retrospective design of the study. Third, sTILs and TIICs results are sensitive to inherent technical difficulties in their determination, such as tumour heterogeneity, partial representation of the tumour in the FFPE blocks, and heterogeneous distribution of sTILs. Finally, our immunoscore requires validation, which is currently ongoing in another cohort of HER2-negative BC.

## 4. Materials and Methods

### 4.1. Patient Population and Outcome Evaluation

We included 118 patients with HER2-negative infiltrating BC undergoing treatment with NACT (anthracycline and taxane regimens) between 2011 and 2017 at the Hospital de la Santa Creu i Sant Pau (HSCSP).

Archived formalin-fixed paraffin-embedded (FFPE) tissue samples from diagnosis were used for sTILs assessment and immunohistochemistry of TIICs. Peripheral blood mononuclear cells (PBMCs) were isolated from whole blood samples of patients collected post-NACT and stored at −80 °C until the determination of systemic biomarkers.

pCR was defined as ypT0/is ypN0 or ypT0 ypN0 after surgery [46]. Clinical response was also evaluated according to the residual cancer burden (RCB) [47] and Miller and Payne (MP) [48,49] systems. Tumours with pCR ypT0/is ypN0, MP-4/5 and MP-A/D, and RCB-0 were considered responders, while those with non-pCR, MP-1/2/3 and MP-B/C, and RCB-I/II and -III were considered non-responders. Overall survival (OS) was defined as the time from diagnosis to last clinical follow-up or death from any cause. Event-free survival (EFS) was defined as the time from the onset of NACT until local or second contralateral tumour, distant progression, death by any cause, or last clinical follow-up. The 2011 St Gallen Consensus [50] and data from Cheang MCU et al. [51] were used to classify molecular subtypes of BC.

### 4.2. sTILs Assessment

sTILs were assessed on hematoxylin-eosin-stained slides from pre-treatment FFPE tissue (N = 118) following the guidelines of the international TIL working group [5], and quantified as the percentage of immune cells in the intratumoural stromal tissue (continuous variable). They were also stratified into high sTILs (>10%) vs. low-sTILs (≤10%), considering the optimal cut-off point obtained from the receiver operating characteristic (ROC) curves that best discriminated pCR in our data. The cut-off point was in line with the literature [4]. Two of the authors (A.G. and C.A.) visually scored the sTILs independently, and discordant results were reviewed until mutual consensus was reached. Intratumoural TILs were also quantified, but they were not included in the analysis.

### 4.3. Immunohistochemistry of TIICs

Serial 5 µm FFPE tissue (N = 100) sections were cut and stained using the Envision method (Dako; Glostrup, Denmark) and CD20 (clone L26), CD4 (clone 4b12), CD8 (clone C8-144B), CD68 (clone KP1), and CD1a (clone 010) antibodies (Dako, Glostrup, Denmark). Two of the authors (AG and CA) independently quantified the number of positive cells in 10 consecutive fields. Discordant results were discussed until a mutual consensus was reached. The cut-off points to stratify into high and low (ROC curves) were calculated.

### 4.4. MDSCs Analysis

After thawing, PBMCs (N = 74) were analysed by flow cytometry using anti-CD11b-PE, anti-CD14-PerCP, and anti-CD11c-APC (Immunotools, Friesoythe, Germany), anti-CD33-APC, and anti-HLA-DR-APCH7 (BD Biosciences, San Jose, CA, USA). Viability was assessed by flow cytometry using the LIVE/DEAD TM Fixable Violet Dead Cell Stain Kit (Thermo Fisher Scientific, Vienna, Austria). MDSC were identified by combining anti-CD14, anti-HLA-DR, anti-CD33, and anti-CD11b. Data analysis was performed using FlowJo (v.10, Ashland, OR, USA). The percentage of cells and the count of cells/µL were determined.

### 4.5. Isolation of Exosomes

PBMCs (N = 74) were thawed to obtain exosomes by serial centrifugation steps, purification, and detection using the ExoStep Plasma kit (Immunostep, Salamanca, Spain), and analysed by flow cytometry with the MACS-Quant Analyser 10 flow cytometer (Miltenyi Biotec, Bergisch Gladbach, Germany).

### 4.6. Statistical Analyses

The Kolmogorov-Smirnov test was used to test the normality of continuous variables. Variables with a normal distribution were shown as mean and standard deviation (SD), and variables with a non-normal distribution were shown as median and interquartile range (IQR). Clinical response was analysed using binary logistic regression when the dependent variable had two categories: pCR (ypT0/is ypN0 or ypT0 ypN0 and non-pCR), MPbreast (MP-4/5 and MP-1/2/3) and Mpaxilla (MP-A/D and MP-B/C). It was analysed using ordinal logistic regression when the dependent variable had more than two ordered categories: RCB (RCB-0, I/II, and –III). The results are expressed as odds ratios (ORs) with 95% confidence intervals (CIs). Survival curves were constructed using the Kaplan-Meier method. Differences were tested using the long-rank test and Cox proportional hazards regression and expressed by hazard ratios (HR) and 95% CIs. sTILs as a continuous variable was analysed using logistic regression for categorical variables and Spearman’s correlation for continuous variables. sTILs as a stratified variable and median TIICs were analysed using the chi-square or Fisher’s exact tests for categorical variables and the non-parametric Mann-Whitney or Kruskal-Wallis tests for continuous variables. In all cases, histological grade (G3 vs. G1/G2), tumour size (T2/T3/T4 vs. T1), lymph node status (N+ vs. N0), and molecular subtype (luminal-A, luminal-B, and TNBC) were included in the models as covariates. Statistical significance was defined as a *p* value < 0.05. Model assumptions were checked using appropriate tests, depending on the models. For the binary and ordered logistic regression models, the assumption of linearity was checked through the Tukey-Pregibon Link test. For the ordered logistic regression, proportional odds assumption was checked through the approximate likelihood-ratio test of proportionality of odds. Finally, the proportional hazard assumption from Cox proportional hazard models was checked by examining Schoenfeld residuals and applying the test of the non-zero slope. Heatmaps and hierarchical clustering of patient data were generated based on the various markers for all patients and for each of the molecular subtypes. The best results were obtained by discarding CD1a from the analysis; therefore, CD4, CD8, CD20, and CD68 markers were chosen for these analyses. The numerical values were transformed as x′ = log(x + 1) to scale them. These analyses were performed with R (v.3.6.3; https://www.R-project.org/, accessed on 18 October 2020) using standard packages, haven packages (R package v.2.2.0; https://CRAN.R-project.org/package=haven, accessed on 18 October 2020), and pheatmap (v.1.0.12; https://CRAN.R-project.org/package=pheatmap, accessed on 18 October 2020) with the default parameter values. To calculate the immunoscore, we performed a multivariate logistic regression using variables with *p* values < 0.2. The regression coefficient was rounded to the nearest integer to calculate the risk score weight, and the ROC curve of the simplified model was evaluated and validated. The optimal cut-off point that defines high and low risk was calculated using the Youden index (J) method (J = Sensitivity + Specificity − 1). Statistical analyses were performed with the STATA 15.1 statistical package, R (v.3.6.3; https://www.R-project.org/, accessed on 19 November 2020), and IBM-SPSS statistical software (v26.0).

## 5. Conclusions

Our data support the role of sTILs in identifying HER2-negative BC patients who are most likely to achieve a complete pathological response with chemotherapy. Our findings also show that an immunological component may be relevant in a subset of patients with HER2-negative tumours, particularly luminal tumours. Finally, once validated in prospective clinical trials, the proposed immunoscore may be useful in clinical decision-making and in treatment escalation or de-escalation.

## Figures and Tables

**Figure 1 ijms-25-02627-f001:**
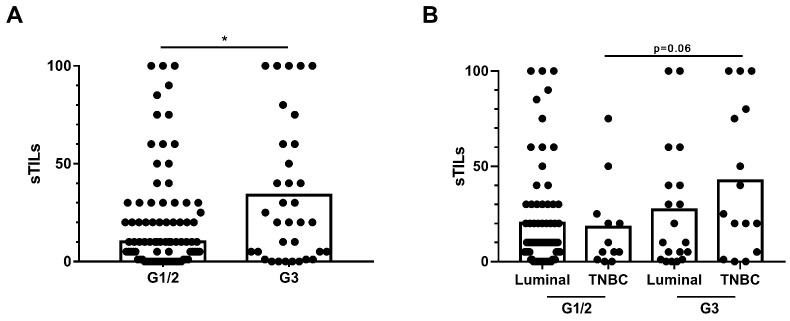
sTILs and clinicopathological features. (**A**) Comparison of sTILs according to histological grade (G). (**B**) Comparison of sTILs according to histological grade and classified as having luminal or triple-negative breast cancer (TNBC) tumours. * *p* < 0.05.

**Figure 2 ijms-25-02627-f002:**
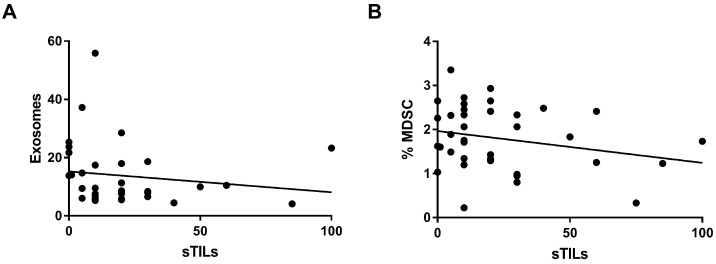
Systemic expression biomarkers. (**A**) Correlation between sTILs and systemic levels of exosomes. (**B**) Correlation between the sTILs with systemic levels of myeloid-derived suppressor cells (MDSC).

**Figure 3 ijms-25-02627-f003:**
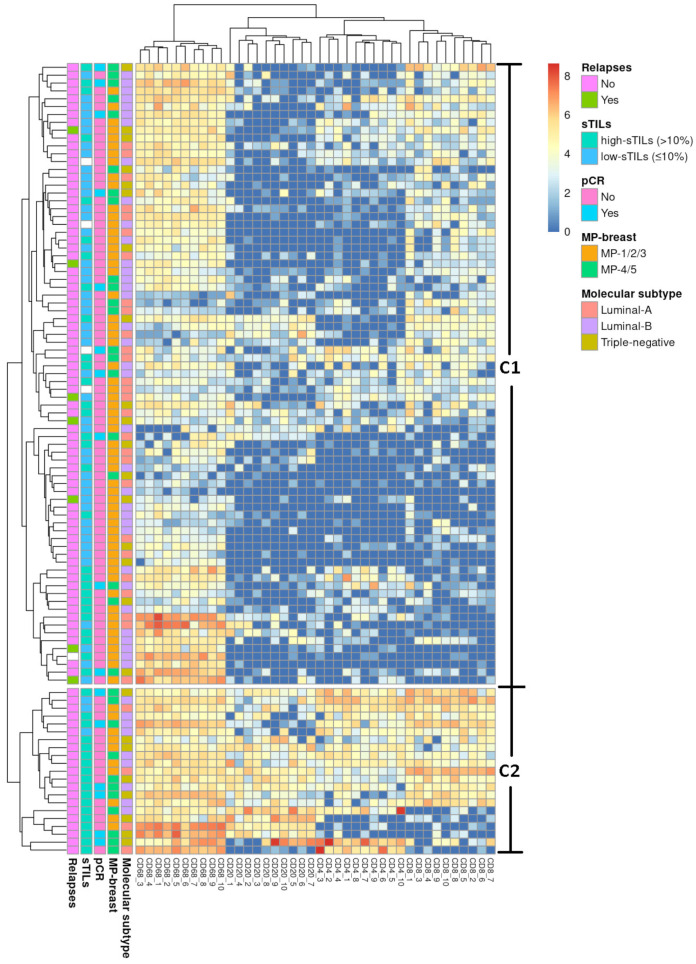
Unsupervised hierarchical cluster of the entire cohort. The heat map represents the unsupervised hierarchical cluster analysis of tumour samples (rows) of the entire cohort and the relative enrichment for specific tumour-infiltrating immune cell subpopulations (CD68, CD20, CD4, and CD8) (columns). It shows two groups of patients: C1 (high risk) and C2 (low-risk).

**Figure 4 ijms-25-02627-f004:**
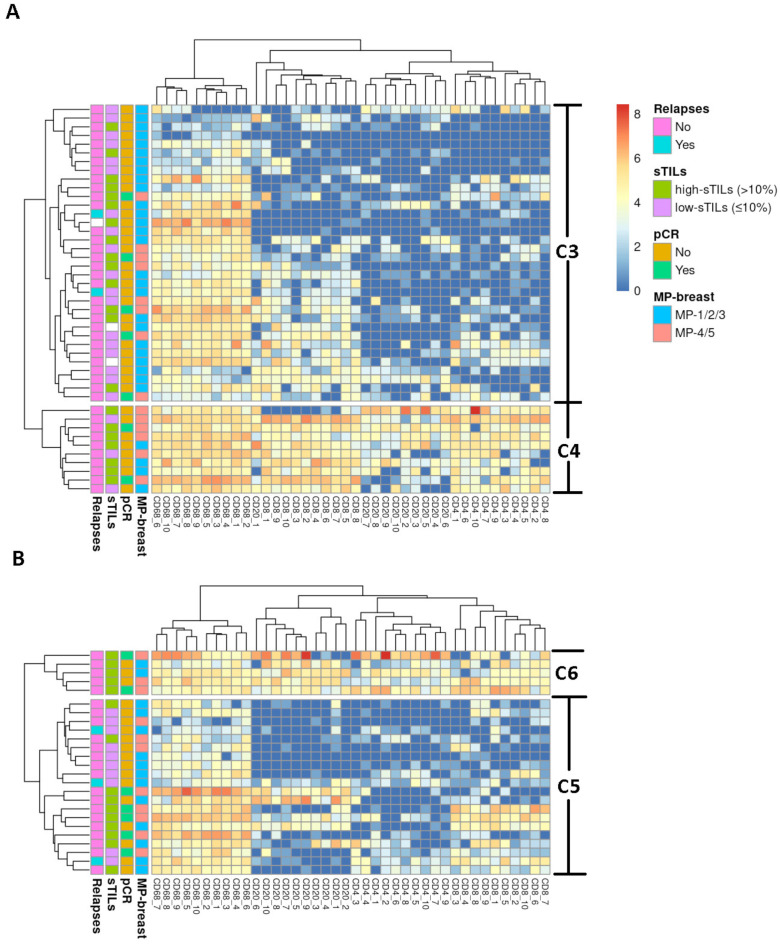
Unsupervised hierarchical cluster of luminal-B and triple-negative (TN) breast cancer (BC) cases. The heat map represents the unsupervised hierarchical cluster analysis of (**A**) luminal-B tumour samples (rows) and the relative enrichment for specific tumour-infiltrating immune cell subpopulations (CD68, CD20, CD4, and CD8) (columns). It shows two groups of patients: C3 (high risk) and C4 (low-risk). (**B**) TNBC tumour samples (rows) and the relative enrichment for specific tumour-infiltrating immune cell subpopulations (CD68, CD20, CD4, and CD8) (columns). It shows two groups of patients: C5 (high risk) and C6 (low-risk).

**Table 1 ijms-25-02627-t001:** Patients’ baseline clinicopathological characteristics (N = 118).

Characteristic	N (%)
**Menopausal status**	
Premenopausal	46 (39.0)
Postmenopausal	67 (56.8)
Perimenopausal	5 (4.2)
**Tumour size**	
T1	11 (9.3)
T2	47 (39.8)
T3	30 (25.4)
T4	30 (25.4)
**Tumour type**	
Ductal	103 (87.3)
Lobular	10 (8.5)
Mixed (ductal and lobular)	2 (1.7)
Other histologic type	3 (2.5)
**Histologic grade**	
G1	9 (7.6)
G2	75 (63.6)
G3	34 (28.8)
**Lymphovascular invasion**	
Negative	98 (83.1)
Positive	20 (16.9)
**Perineural invasion**	
Negative	108 (91.5)
Positive	10 (8.5)
**ki-67 index**	
<14%	44 (37.3)
≥14%	74 (62.7)
**Clinical N-stage**	
N0	50 (42.4)
N+	68 (57.6)
**Estrogen receptor status**	
Positive	90 (76.3)
Negative	28 (23.7)
**Progesterone receptor status**	
Positive	74 (62.7)
Negative	44 (37.3)
**Pathological complete response**	
pCR	16 (13.6)
Non-pCR	102 (86.4)
**MP-breast grading system**	
1	11 (9.3)
2	20 (16.9)
3	53 (44.9)
4	17 (14.4)
5	17 (14.4)
**MP-axilla grading system**	
A	44 (37.3)
B	24 (20.3)
C	35 (29.7)
D	12 (10.2)
NA	3 (2.5)
**RCB class**	
0	16 (13.6)
I	10 (8.5)
II	60 (50.8)
III	32 (27.1)

pCR: pathological complete response, RCB: residual cancer burden, MP: Miller and Payne, sTILs: stromal tumour-infiltrating lymphocytes, NA: not available.

**Table 2 ijms-25-02627-t002:** Univariate and multivariate analyses of sTILs and their association with pathological response.

	sTILs Continuous Variable				sTILs Stratified Variable			
	Univariate Analyses		Multivariate Analyses *		Univariate Analyses		Multivariate Analyses *	
	OR (95% CI )	*p*	OR (95% CI )	*p*	OR (95% CI )	*p*	OR (95% CI )	*p*
pCR	1.22 [1.05; 1.42]	**0.010**	1.20 [1.00; 1.43]	**0.046**	4.46 [1.35; 14.80]	**0.011**	3.73 [1.01; 13.74]	**0.048**
MPbreast	1.18 [1.04; 1.34]	**0.013**	1.17 [1.02; 1.35]	**0.027**	2.62 [1.16; 5.96]	**0.022**	2.45 [1.02; 5.88]	**0.045**
MPaxilla	1.05 [0.93; 1.19]	0.458			1.46 [0.70; 3.05]	0.324		
RCB	1.11 [0.98; 1.26]	0.110			1.43 [0.69; 2.95]	0.333		

sTILs: stromal tumour-infiltrating lymphocytes, pCR: pathological complete response, MP: Miller and Payne, RCB: residual cancer burden, OR: odds ratio, CIs: confidence intervals. sTILs stratified variable: high-TILs (>10%) vs. low-TILs (≤10%); Responders: pCR ypT0/is ypN0, MP-4/5 and MP-A/D, and RCB 0; Non-responders: non-pCR, MP-1/2/3 and MP-B/C, and RCB I/II and III. * The multivariate analyses included histological grade (G3 vs. G1/G2), tumour size (T2/T3/T4 vs. T1), lymph node status (N+ vs. N0), and molecular subtype (luminal-A, luminal-B, TNBC) as covariates. The statistical parameters shown in the table refer to sTILs (continuous and stratified variables). Statistically significant *p*-values are marked in bold.

**Table 3 ijms-25-02627-t003:** Univariate and multivariate analyses of sTILs and their association with survival.

	sTILs Continuous Variable		sTILs Stratified Variable			
	Univariate Analyses		Univariate Analyses		Multivariate Analyses *	
	HR (95% CI)	*p*	HR (95% CI)	*p*	HR (95% CI)	*p*
EFS	1.53 [0.91; 2.57]	0.109	7.53 [0.95; 59.5]	0.055	11.28 [1.33; 96.03]	**0.027**
OS	1.25 [0.75; 2.07]	0.388	2.39 [0.25; 23.0]	0.450		

sTILs: stromal tumour-infiltrating lymphocytes, EFS: event-free survival, OS: overall survival, HR: hazard ratio, CIs: confidence intervals. TILs stratified variables: high-TILs (>10%) vs. low-TILs (≤10%). * The multivariate analyses included histological grade (G3 vs. G1/G2), tumour size (T2/T3/T4 vs. T1), lymph node status (N+ vs. N0), and molecular subtype (luminal-A, luminal-B, TNBC) as covariates. Statistically significant *p*-values are marked in bold.

**Table 4 ijms-25-02627-t004:** Univariate analyses of sTILs and their association with clusters.

	sTILs Continuous Variable		sTILs Stratified Variable	
	OR (95% CI)	*p*	OR (95% CI)	*p*
Entire cohort (C1 vs. C2)	0.71 [0.60–0.84]	**<0.001**	0.11 [0.03–0.39]	**<0.001**
Luminal B (C3 vs. C4)	0.75 [0.57–0.97]	**0.028**	0.33 [0.07; 1.53]	0.172
TNBC (C5 vs. C6)	0.79 [0.60; 1.05]	0.104	0.00 [0.00; NA]	0.057

sTILs: stromal tumour-infiltrating lymphocytes, OR: odds ratio, CIs: confidence intervals. TILs stratified variables: high-TILs (>10%) vs. low-TILs (≤10%). The statistical parameters shown in the table refer to sTILs (continuous and stratified variables). Statistically significant *p*-values are marked in bold.

**Table 5 ijms-25-02627-t005:** Univariate and multivariate analyses of the clusters identified by unsupervised hierarchical clustering analysis and their association with pathological responses.

	Entire Cohort (C1 vs. C2)				Luminal-B (C3 vs. C4)			
	Univariate Analyses		Multivariate Analyses *		Univariate Analyses		Multivariate Analyses *	
	OR (95% CI)	*p*	OR (95% CI)	*p*	OR (95% CI)	*p*	OR (95% CI)	*p*
MPbreast	0.31 [0.11; 0.83]	**0.024**	0.36 [0.13; 1.03]	0.057	0.21 [0.05; 0.91]	**0.046**	0.15 [0.03–0.77]	**0.023**
MPaxilla	0.56 [0.20; 1.53]	0.265			0.18 [0.03; 0.95]	**0.039**	0.10 [0.01–0.70]	**0.020**
pCR	0.36 [0.11; 0.15]	0.103			0.69 [0.11; 4.24]	0.691		
RCB	0.46 [0.18; 1.20]	0.113			0.40 [0.10; 1.58]	0.193		

MP: Miller and Payne, pCR: pathological complete response, RCB: residual cancer burden, OR: odds ratio, CIs: confidence intervals. * The multivariate analyses included histological grade (G3 vs. G1/G2), tumour size (T2/T3/T4 vs. T1), lymph node status (N+ vs. N0) for Luminal-B, and also molecular subtype (luminal-A, luminal-B, TNBC) for the entire cohort. Statistically significant *p*-values are marked in bold.

## Data Availability

The data presented in this study are available from the corresponding authors. The data are not publicly available due to ethical committee regulations.

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
