# Peer review of "Clinical Relevance of Tumour-Infiltrating Immune Cells in HER2-Negative Breast Cancer Treated with Neoadjuvant Therapy"

_ijms, 2024, doi:10.3390/ijms25052627_

Round 1

Reviewer 1 Report

Comments and Suggestions for Authors

This article adheres to the format of an original article and focuses on an extremely current theme, providing a comprehensive and detailed analysis of the issue. Structured in accordance with the standards of an original article, the presentation emphasizes both rigorous form and the solid foundation of the authors' research. The authors detail the methodology, demonstrating a meticulous approach to research, and the study's foundation is solid. The study is well-conducted, with carefully chosen criteria, and the obtained data are presented clearly and concisely, supported by relevant graphs and tables. Results are analyzed in-depth, emphasizing key conclusions and their significance in the chosen theme's context. Discussions cover potential dilemmas regarding the issue, showcasing the authors' critical and profound understanding. The bibliography is current and well-represented, contributing to the overall strength of the article.

In conclusion, I congratulate the authors for the rigor and quality of the scientific material.

1. What do you think could be the technical factors hindering this type of evaluation, the qualitative determination of the association of sTILs with pathological response and survival? Are there difficulties in assessing and defining tumor boundaries, resulting in an impact on sTILs evaluation?

2. The work is an original one, supporting a modern method. Clearly, being a research study, it comes with a set of possible controversies or difficulties: technical issues (such as slide quality, heterogeneous distribution of sTILs, etc.), the clinical significance of the obtained data, etc. The authors cannot, and understandably so given the limitations of the material and editorial format, comprehensively cover all aspects of the chosen subject.

3. There are few works on the studied and presented subject by the authors.

4. Few changes can be made at this point. Remote evaluations, as well as other studies that will emerge on this issue, will help in a correct assessment over time of this method.

5. I have already stated what needed to be said here.

6. The bibliography is current and well-represented, contributing to the overall strength of the article.

7. The tables and graphics used are part of the applied evaluation of the study, of the obtained results, and are necessary to support this method.

Reviewer 2 Report

Comments and Suggestions for Authors

The authors developed a study with a very important proposal for the breast cancer clinic, which is the prediction of response to neoadjuvant treatment. However, I raise some serious points:

The methodology does not make clear when the biological material sample was collected for systemic markers, but I understand that it was after NAC, since, for immunohistochemistry, the authors report that it was pre-treatment, but do not report it for peripheral blood. Furthermore, if they were frozen, the authors did not report this, which is important as the analysis of fresh and frozen samples has significant differences. Searching the internet, I found another study by the authors (DOI: 10.3390/cancers13246167), with the same Institutional Review Board approval (IIBSP-SPA-2016-76), which reports that the blood was obtained post-NAC. Since the authors' objective is to predict response to neoadjuvant therapy, this creates a serious methodological flaw. Therefore, part of the authors' discussion (lines 298 to 303) does not describe that the peripheral blood sample is not prior to NAC.

The authors describe many results as "trending toward significance." If the authors used a p-value <0.05 as a significance criterion (alpha), the frequent use of this expression leads readers to the wrong conclusion that the observed phenomenon was "almost" significant, along with other conjectures, such as " if the sample n were larger, there would be significance", which hinders the appropriate conclusion on the part of readers (conclusion bias).

The authors do not present the distribution analysis of the data carried out for each variable. Even so, it was assumed that the median and standard deviation are the best descriptors of central tendency and data dispersion for TIICs. There is already a contradiction in this aspect since for the median, minimum-maximum or IQR is used for dispersion; SD is for mean (symmetric/Normal distributions). Such analyses are fundamental for the appropriate choice of statistical tests/methods.

The authors do not describe how they tested the proportionality of continuous variables, such as sTIL, for logistic or Cox regression.

The authors assume ordinal logistic regression to test the dependents RCB and MP, but do not report a parallel regression test that confirms that such regression is adequate; If not, binary or multinomial should be used.

Other minor points:

Based on the reference provided by the authors (DOI: 10.1016/S0960-9776(03)00106-1) and other articles (DOI: 10.21037/gs-21-608; 10.3389/fonc.2021.735670), the Miller-Paine system deals with the degree of response (G1-G5) in the tumor bed (T). Where did the authors obtain an armpit MP system?

The authors do not describe hierarchical clustering or its parameters in the "Statistical analysis" subsection.

The presentation of results is often confusing. For example, in lines 202 and 203, it says:

"and 202 with MDSCs (r = -0.277; p = 0.039 and median MDSC high-sTILs: 1.4 (1.1-2.2) vs low-sTILs: 203 2.0 (1.5-2.5); p = 0.031)".

Combining results from different statistical methods in the same sentence makes it difficult for readers to understand. Such results can be organized in tables and separated by methodologies.

In multivariable regressions, at least in the footer, it must be described which variables were included in the final model.

The tables do not present the main variable tested, which is only in the header, without reporting OR/HR (CI95%) and p-value.

The tables are confusing. They should be fragmented instead of aggregated as the actors did.

Round 2

Reviewer 2 Report

Comments and Suggestions for Authors

The authors did an excellent job of answering several questions. However, I still raise a few:

1. Regarding the proportionality of continuous variables for logistic or Cox regression models, the authors did not answer the question. I apologize it is the proportionality of risks for Cox regression and linearity for logistic regression. I was not referring to correlations, but rather to prerequisites. For logistic regression, there must be a relationship between the logit of the outcome and each predictor variable, especially when it comes to continuous variables. For Cox regression, as the name suggests (Cox proportional hazards model), there must be proportionality, as well as checking for non-linearity. The authors did not answer this question.

2. Similarly to the previous point, the authors did not answer how they checked whether the ordinal regression performed meets one of the requirements: proportional odds. If there were no proportional chances between the classifiers (categories of the ordinal variable), there is more than one model, and multinomial regression should be used instead. The authors did not answer this question.

3. Regarding the tables, the authors made corrections that greatly help with interpretation. However, they have not yet included the ORs/HRs of the sTILs variables, continuous or categorical/stratified, in the tables, which is the main result of the study. These data are still only written in full in the manuscript, making it difficult to interpret the results when only looking at the table.

Round 3

Reviewer 2 Report

Comments and Suggestions for Authors

The authors were able to answer the main questions. However, the results of the univariable and multivariable analyses of sTIL are not yet available in the tables (OR, [95%CI] and p-value).

Taking Table S1 as an example, the authors presented such parameters for the variables Tumor size (1.09 [0.91;1.31] 0.354), Histological grade (0.86 [0.76;0.98] 0.023), Lymphovascular invasion (1.01 [0.86;1.20] 0.882), Perineural invasion (0.99 [0.80;1.23] 0.945), Ki-67 index (0.88 [0.76;1.02] 0.088), Clinical N-stage (0.97 [0.86;1.10] 0.688), and Hormonal receptor-status (0.90 [0.79 ;1.03] 0.129), but not for sTILs, which is the main variable of the study.

Author Response

Thank you. In tables 2, 4 and S1 all the ORs, [95%CI] and p-values are referred to sTILS as continuous and stratified variables.

To avoid further misunderstanding, we have now added an explanation in the foot of tables 2, 4 and S1. Now it reads: “The statistical parameters shown in the table refer to sTILs (continuous and stratified variables)”.